EMBO
Molecular Medicine

# The respiratory syncytial virus (RSV) prefusion F-protein functional antibody repertoire in adult healthy donors

Emanuele Andreano[1,2,†] , Ida Paciello[2,†], Monia Bardelli[2], Simona Tavarini[2], Chiara Sammicheli[2], Elisabetta Frigimelica[2], Silvia Guidotti[2], Giulia Torricelli[2], Marco Biancucci[3], Ugo D'Oro[2], Sumana Chandramouli[3,‡], Matthew J Bottomley[3], Rino Rappuoli[2,4,5], Oretta Finco[2] & Francesca Buricchi[2,*]

## Abstract

Respiratory syncytial virus (RSV) is the leading cause of death from lower respiratory tract infection in infants and children, and is responsible for considerable morbidity and mortality in older adults. Vaccines for pregnant women and elderly which are in phase III clinical studies target people with pre-existing natural immunity against RSV. To investigate the background immunity which will be impacted by vaccination, we single cell-sorted human memory B cells and dissected functional and genetic features of neutralizing antibodies (nAbs) induced by natural infection. Most nAbs recognized both the prefusion and postfusion conformations of the RSV F-protein (cross-binders) while a smaller fraction bound exclusively to the prefusion conformation. Cross-binder nAbs used a wide array of gene rearrangements, while preF-binder nAbs derived mostly from the expansion of B-cell clonotypes from the IGHV1 germline. This latter class of nAbs recognizes an epitope located between Site Ø, Site II, and Site V on the F-protein, identifying an important site of pathogen vulnerability.

**Keywords** functional antibody repertoire; human monoclonal antibodies; RSV; vaccine development
**Subject Categories** Immunology; Microbiology, Virology & Host Pathogen Interaction

## Introduction

Human respiratory syncytial virus (RSV) is a RNA virus of the *Orthopneumovirus* genus (*Pneumoviridae* family) (Rima *et al*, 2017). The virus causes acute lower respiratory infection (ALRI) especially in children under the age of 5 and older adults (Nair *et al*, 2010). Approximately, two-thirds of infants are exposed to RSV at least once during the first year of life, and approximately 90% experience RSV infection one or more times by 2 years of age (Collins & Graham, 2008). Approximately, 33.1 million episodes of RSV-mediated ALRI and 94,600–149,400 deaths were estimated globally in 2015 (Shi *et al*, 2017). These observations indicate that virtually all adult humans have suffered repeated infections during their lifetimes and developed a pathogen-specific immune response (Hall *et al*, 1991; Graham *et al*, 2015; Gilman *et al*, 2016).

Currently, the humanized monoclonal antibody palivizumab (MedImmune, Synagis) is the only prophylactic tool approved for use to protect high-risk infants against RSV, and no vaccine is available against this virus (Simoes, 1999; Company ME, 2003; COID, 2009; Hurwitz, 2011; AAP, 2014). The lack of an approved vaccine represents an important unmet medical need.

The RSV genome encodes for 11 proteins of which the type I integral membrane fusion protein (F-protein) plays a pivotal role in the pathogenesis of RSV by mediating the fusion between the viral and the host cell membrane (Barretto *et al*, 2003; Collins 2007; Levine *et al*, 1987; Teng & Collins, 2002). The F-protein presents two different conformations, a lollipop-shaped prefusion (preF), present on the virus surface before virus–cell interaction, and a crutch-shaped postfusion (postF) state which is acquired following the fusion between the virus and cell membrane or by unknown mechanisms that spontaneously initiate the rearrangement from the highly

1 Department of Life Sciences, University of Siena, Siena, Italy
2 GSK Vaccines, Siena, Italy
3 GSK Vaccines, Rockville, MD, USA
4 Faculty of Medicine, Imperial College, London, UK
5 Monoclonal Antibody Discovery (MAD) Lab, Fondazione Toscana Life Sciences, Siena, Italy
*Corresponding author. Tel: +39 3454438292; E-mail: francesca.x.buricchi@gsk.com
†Present address: Monoclonal Antibody Discovery (MAD) Lab, Fondazione Toscana Life Sciences, Siena, Italy
‡Present address: Moderna Therapeutics Inc, Cambridge, MA, USA

metastable preF into the energetically favorable postF conformation (Mejias *et al*, 2017). The two forms are antigenically distinct, and both are considered as potential vaccine candidates (Swanson *et al*, 2011; Graham, 2019).

However, the preF form has been shown to induce the majority of highly neutralizing antibodies following natural infection or immunization, making this the antigen of choice for vaccine development (Calder *et al*, 2000; Swanson *et al*, 2011; Flynn *et al*, 2016; Falloon *et al*, 2017; Sastry *et al*, 2017; Steff *et al*, 2017; Rossey *et al*, 2018). In fact, the two most F-protein immunogenic sites, named Site Ø and Site V, are only present on the preF conformation, while less immunogenic sites (Site I, Site II, Site III, and Site IV) are shared between the preF and postF forms of this antigen (Graham, 2019). In order to better understand the natural response to RSV, we studied the neutralizing antibody repertoire against the preF form of the antigen in healthy adult donors. Shedding light on the antibody functional repertoire against the preF conformation may help to better understand those features that characterize an effective response and which germline expansions would be most beneficial following vaccination. In fact, understanding the nature of the protective antibody response following RSV natural infection is of fundamental importance for the development of a RSV vaccine since their response and effectiveness will build on their ability to engage and evolve pre-existing B-cell immunity.

Applying elements of the Reverse Vaccinology 2.0 methodology, which uses knowledge of the human immune response developed following natural infection and/or vaccination for antigen discovery and vaccine development (Rappuoli *et al*, 2016), we isolated and cultured over 1,200 IgG$^+$ preF-specific single memory B cells (MBCs) from four healthy adults and studied the functional and genetic properties of neutralizing antibodies produced by the single-cell clones. These analyses allowed us to characterize clonally expanded neutralizing antibody (nAb) families and to shed light on the predominant RSV F-specific heavy and light chain gene rearrangements. These results describe functionally and genetically the immunological repertoire elicited by RSV natural infection in healthy adults, identifying the characteristics of an effective antibody response and providing key translational information that support the development of vaccines based on the preF antigen (Walsh *et al*, 2004; Falsey *et al*, 2005; Paes *et al*, 2011; Resch, 2014).

# Results

## Identification of naturally induced preF-binding antibodies from healthy adult donor MBCs

To retrieve RSV preF-binding antibodies, B cells from peripheral blood mononuclear cells (PBMCs) of four adult healthy donors were stained with fluorescent-labeled preF trimer. CD19$^+$, IgA$^-$, IgD$^-$, IgM$^-$, and preF$^+$ cells were single cell-sorted into 384-well plates in order to isolate only IgG$^+$ memory B cells (MBCs) binding to the F-protein in its preF state (Fig EV1). More than 1,200 IgG$^+$ preF-binding MBCs were successfully retrieved from these subjects. The frequencies of CD19$^+$, IgA$^-$, IgD$^-$, IgM$^-$, and preF$^+$ cells in the four donors ranged from 0.01 to 0.05% (Fig 1A). Single cell-sorted preF$^+$ MBCs were incubated over a layer of 3T3-CD40L feeder cells in the presence of IL-2 and IL-21 stimuli for 2 weeks (Huang *et al*, 2013).

Subsequently, the presence of IgG in each single B-cell culture supernatant was tested using a human IgG-specific ELISA. Out of 1,210 MBCs, 462 produced detectable amounts of IgG in the culture supernatant, showing an average clonal efficiency of 38.3% (Fig 1B). Secreted IgG concentrations were extremely variable, ranging from < 100 ng/ml to over 32 μg/ml (Fig EV2), likely reflecting the diverse level of expansion among single clones.

As a first step for the characterization of isolated mAbs, their binding to RSV preF was tested. The binding assay was performed using the Gyrolab microfluidic system, which allows high-throughput and highly sensitive screening of antibodies against the antigen of choice. From the 462 MBC supernatants with confirmed IgG$^+$ production, a total of 356 contained preF-binding IgG, corresponding to an average clonal specificity of 68.1% (Fig 1C).

## Functional characterization of naturally induced antibodies from healthy adult donor preF-binding MBCs

All 356 supernatants containing preF-binding IgG were screened for their capacity to neutralize RSV *in vitro* by plaque reduction neutralization assay (PRNA). From the 356 mAbs analyzed in this step, a total of 135 neutralizing antibodies (nAbs) were identified (21, 14, 17, and 83 for Donors 503, 203, 103, and 003 respectively) (Fig 2A). The screening was aimed only to qualitatively assess these mAbs for their neutralization activity and not to compare them for neutralization potency. Indeed, since supernatant IgG concentration was highly variable and sample availability following single cell sorting and 2-week incubation was extremely limited (~50 μl/well), it was not feasible to test all mAbs at the same concentration. Hence, all mAbs were tested at the same supernatant dilution (1:5). No correlation between antibody concentration and neutralization (positivity in PRNA assay) was observed; for example, some mAbs at concentrations as low as 100 ng/ml were still capable of neutralizing the virus (Fig EV2).

To better characterize the isolated nAbs, a competition assay was performed using the Gyrolab microfluidic system. One hundred and twenty-nine MBC supernatants were tested (6 of the 135 total were not tested due to insufficient reagent volume) against three well-characterized neutralizing Fabs (D25, motavizumab (Mota) and 101F (McLellan *et al*, 2010a; McLellan *et al*, 2010b; McLellan *et al*, 2013) with the aim of identifying the epitopes to which the new nAbs could bind. The majority of nAbs isolated from Donor 503, 203, and 003 recognized known epitope regions, and many competed with the highly neutralizing preF-specific D25 and/or the cross-binding Mota, which recognize Site Ø and Site II, respectively (McLellan *et al*, 2010b; McLellan *et al*, 2013) (Fig 2B). These data further confirm that nAbs targeting preF often target Site Ø and Site II regions.

Interestingly, the binding prevalence toward the specific epitope regions observed in the three donors 503, 203, and 003 was not observed in the fourth donor 103; i.e., we did not identify nAbs that competed with both D25 and Mota from Donor 103. In addition, 33% of nAbs isolated from Donor 103 did not compete with any of the three Fabs tested in our assay. This percentage is 5- to 30-fold higher than Donors 503, 203, and 003 that showed only 5, 7, and 1%, respectively.

Another noteworthy antibody sub-population is comprised of nAbs that compete exclusively with D25, given that most of the highly potent RSV nAbs described to date recognize the corresponding Site Ø epitope on preF (Rossey *et al*, 2018). Although nAbs from Donor 103 did not compete with both D25 and Mota, antibodies that

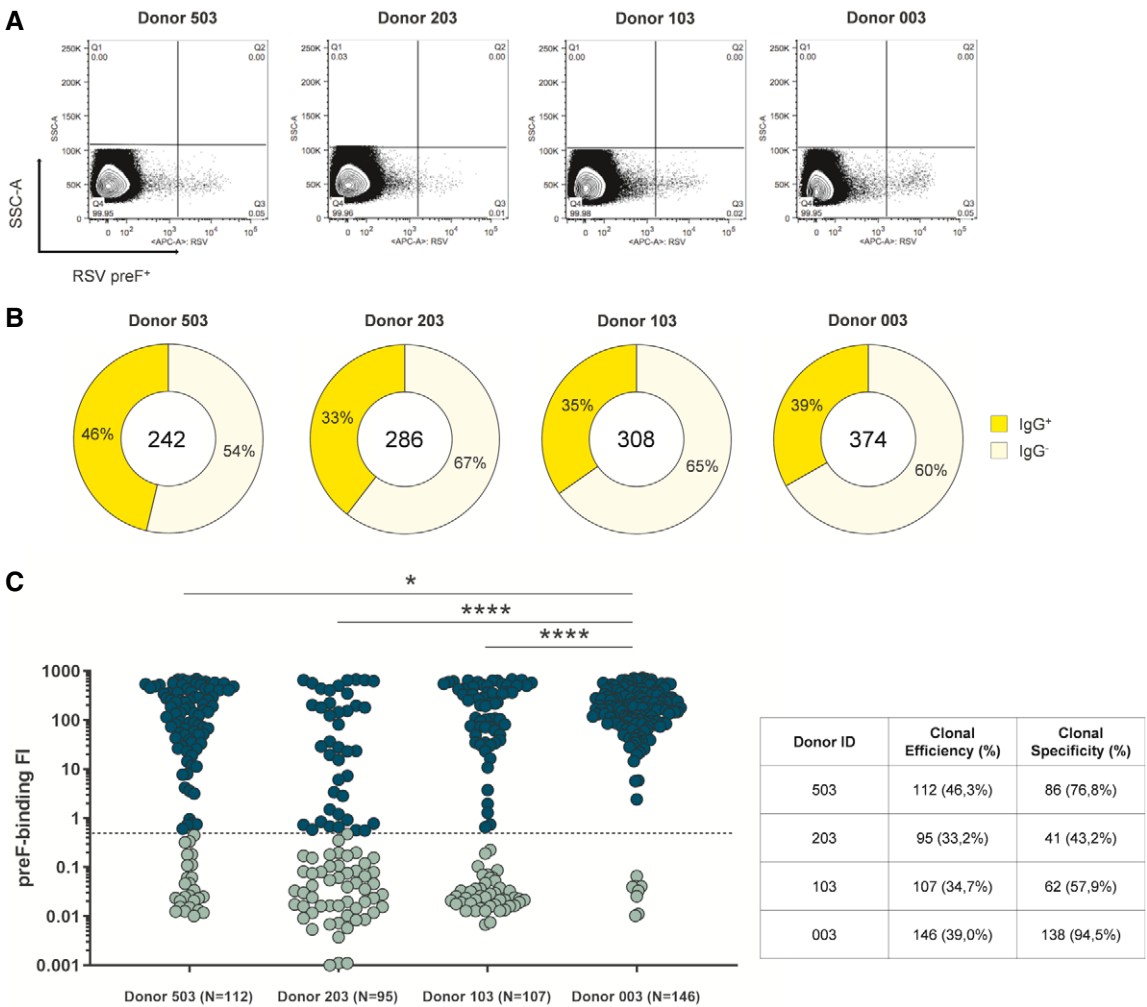

**Figure 1. Isolation of RSV preF-binder mAbs naturally produced from MBCs.**

MBCs IgG⁺ RSV preF binder were retrieved from four adult healthy donors.

A  Sorting plots show in quadrant 3 (Q3, bottom right) RSV preF fluorescent-activated cells.

B  Donut charts show in the center the total number of cells sorted per subject. In dark yellow are shown percentages of supernatant presenting detectable IgG (clonal efficiency) which, starting from left to right, were 46.3% (N = 112), 33.2% (N = 95), 34.7% (N = 107), and 39.0% (N = 146), respectively, averaging 38.3% overall. In pale yellow are shown percentages of supernatant presenting undetectable IgG.

C  The graph shows supernatant IgG⁺ tested for binding to the F-protein in its preF conformation. Threshold of positivity has been set as three times the value of IgG⁻ supernatant. Dark green dots are IgG preF binders while light green dots are non-binder IgGs. The table summarizes numbers and percentages of IgG preF binders (clonal specificity). The *P*-value was calculated with the nonparametric One-way ANOVA test, and significance is shown as * ($P \leq 0.05$), ** ($P \leq 0.01$), *** ($P \leq 0.001$), and **** ($P \leq 0.0001$).

competed solely with D25 were observed (7%), even if in a noticeably reduced proportion as compared to Donor 503 and Donor 003 (24 and 21%, respectively) (Fig 2B). Interestingly, the nAbs panel from Donor 203, which possessed the second highest percentage of nAbs (Fig 2A), did not present any D25-only competing nAbs, suggesting that an epitope region in-between Site Ø and Site II could also be an important contributor to virus neutralization (Fig 2B).

**Binding preference of nAbs for PreF versus PostF conformation**

Following mAb competition assessment, 82 isolated nAbs (47 of the 129 total nAbs were not tested due to insufficient reagent volume) were selected for testing of binding to preF and postF conformations to better characterize their antigen specificity, using a Gyrolab fluorescence intensity (FI)-based assay. Although nAbs were originally isolated by using the preF antigen to stain MBCs, it was interesting to observe that the majority of them were cross-binders, hence capable of binding epitopes shared between preF and postF (Fig 2C). Interestingly, Donor 003 that showed a significantly higher amount of nAbs (Fig 2A) was the only donor with the majority of nAbs being preF-specific (Fig 2C) further confirming the importance of this antigen conformation in eliciting neutralizing antibodies. However, despite their ability to recognize both molecules, only 4 of 82

cross-binding nAbs displayed greater binding to the postF protein, indicating an overall preferential binding toward the PreF conformation (Fig 2D). In summary, as expected, most neutralizing mAbs were found to exhibit a binding preference for the PreF protein, which is likely to be the major species present on the virion surface.

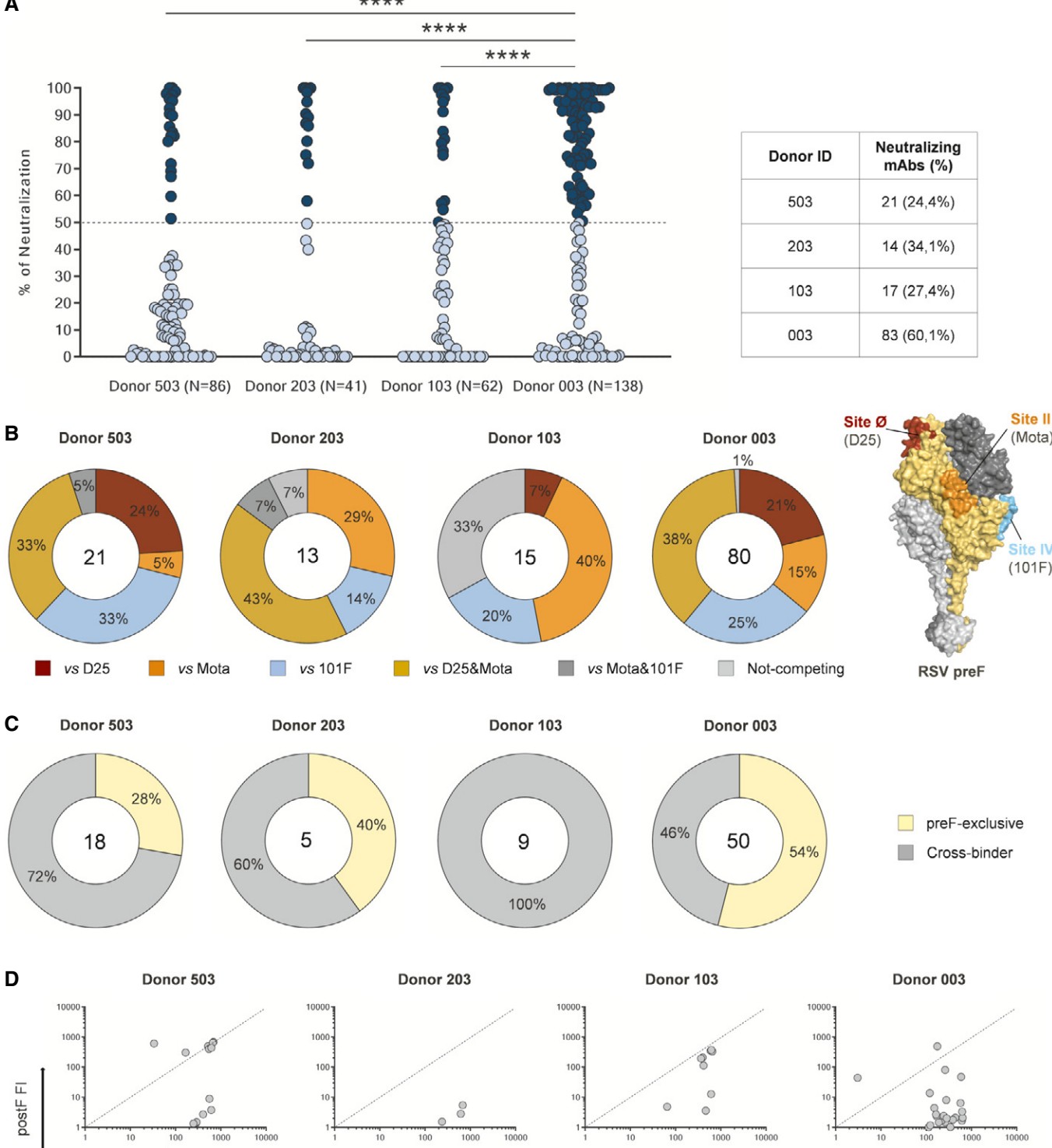

Figure 2.

**Figure 2.  Functional characterization of identified RSV F-protein-specific nAbs.**

A   The graph shows RSV preF-binder mAbs tested by PRNA to assess neutralization efficiency. For high-throughput and qualitative screening, mAbs were tested at a single point dilution (1:5). Threshold of positivity has been set as 50% reduction of syncytia compared to the virus tested alone on Vero feeder cells. Dark blue dots show neutralizing mAbs while light blue dots represent non-neutralizing mAbs. The table shows numbers and percentages of nAbs identified for each adult healthy donor. The *P*-value was calculated with the nonparametric One-way ANOVA test, and significance is shown as * ($P \leq 0.05$), ** ($P \leq 0.01$), *** ($P \leq 0.001$), and **** ($P \leq 0.0001$).

B   Competition assay has been performed to speculate on the epitopic region recognized by preF-binder nAbs. Known neutralizing Fabs have been used to test nAbs for binding toward sites Ø (D25), II (Mota), and IV (101F). Donut charts show in the center the total number of nAbs.

C   Identified nAbs were assessed for binding against both preF and postF to assess their binding specificity. Yellow and gray slices represent the percentage of nAb exclusive preF binders and pre/postF cross-binders, respectively. In the center of the donut charts is shown the total number of nAbs tested for binding against both conformations of the F-protein.

D   The cross-binder nAbs binding fluorescence intensity (FI) values against either preF (X-axis) or postF (Y-axis) were plotted to obtain the ratio of binding and understand the preferred targeted conformation of the F antigen.

## IGHV1 gene family is preferentially expanded in the F-specific functional antibody repertoire

Following functional characterization of F-protein-specific nAbs, their sequences were retrieved using single-cell MBC PCR methods and were analyzed to gain insights into the heavy and light chain gene rearrangements specific to the functional repertoire of healthy adult donors. One hundred and fifteen sequences out of the 135 identified nAbs were retrieved by PCR (overall average PCR efficiency 85.2%) and 86 of these were successfully sequenced and analyzed (overall average sequencing efficiency 74.8%) (Fig EV3).

The functional repertoire analyses of the 86 Ig gene sequences revealed that RSV F-specific nAbs use several heavy chain gene families, of which the IGHV1 is the most frequently found (Fig 3A). In particular, two predominant heavy chain V genes were identified, IGHV1-18 and IGHV1-69. Furthermore, IGHV1-18 preferentially paired with the J gene IGHJ4-1, while IGHV1-69 paired principally with two different J genes, IGHJ3-1 and IGHJ4-1. These heavy chain V-J gene rearrangements represented 14.5, 8.5, and 9.4% of the whole repertoire, respectively, and were mostly observed in Donor 003 (Fig 3A) which showed the highest amount of preF-specific nAbs among donors (Fig 2C). In addition, these rearrangements present a very similar heavy chain complementarity determining region 3 (H-CDR3) length, ranging from 13 to 16 amino acid (aa) residues (average length 14.9 aa), compared to non-predominant rearrangements which exhibited more variable and slightly longer H-CDR3 loops (average length 16.6 aa) (Fig 3B). The V-gene somatic hypermutation levels of predominant IGHV1-69 gene-derived nAbs showed a significantly higher mutation frequency compared to the non-predominant nAbs. This is not the case for IGHV1-18;IGHJ4-1 gene-derived nAbs (Fig 3C). Finally, following the analyses of the paired heavy and light chain repertoire, it is interesting to observe that all the IGHV1-18; IGHJ4-1 paired with IGKV2-30 (particularly with IGKV2-30; IGKJ1-1 or IGKJ2-1) while IGHV1-69; IGHJ3-1 paired almost exclusively with IGKV3-20; IGKJ1-1 (Fig EV4).

## Competitor nAbs show preferential heavy chain V-J gene rearrangements

In this study, we aimed to understand whether antibodies that compete with known neutralizing Fabs (D25, Mota, and 101F) present specific V-J gene signatures. nAbs competing exclusively with D25, Mota, 101F or with both D25 and Mota, showed a preferential use of different heavy chain V genes, namely IGHV1/3, IGHV4, IGHV3, and IGHV1, respectively (Fig 4A). In detail, nAbs competing exclusively with D25 (Fig 4A, top left) mainly use the predominant IGHV1-69;IGHJ4-1 ($N = 6$; 31.6%) and non-predominant IGHV3-15; IGHV4-1 ($N = 6$; 31.6%) heavy chain V-J gene rearrangements. nAbs competing exclusively with Mota (top right) use mainly the IGHV4-34;IGHJ3-1 ($N = 6$; 31.6%) rearrangement. 101F competing nAbs (bottom right) were more variable in terms of V-J gene rearrangements usage, and the IGHV3-13;IGHJ6-1 is the most utilized ($N = 4$; 14.8%). Finally, nAbs competing with both D25 and Mota (herein referred to as "D25&Mota competing nAbs") (bottom left) clearly showed the IGHV1-18;IGHJ4-1 as predominant heavy chain V-J gene rearrangement ($N = 14$; 36.8%) (Fig 4A).

Furthermore, the amino acid sequences of H-CDR3 loops of nAbs showing predominant gene rearrangements were grouped based on their competition profile and were analyzed (Fig 4B). The aim was to identify possible conserved motifs within these CDR loop regions. To avoid biases, the analysis was performed only for sequence groups which met the following criteria: originating from different subjects and containing at least 5 members. While no conserved motifs were observed in the H-CDR3 of IGHV1-69;IGHJ4-1 D25 competing nAbs ($N = 6$), a sequence motif was identified in IGHV1-18;IGHJ4-1 gene-derived D25&Mota competing nAbs ($N = 14$). Indeed, two consecutive alanines were observed in positions 10 and 11 of nearly all the H-CDR3 which tend to be flanked, on both sides, by amino acids presenting electrostatically charged side chains (Fig 4B). The same motif, which has two alanines at the center of the H-CDR3, was also observed in two previously identified IGHV1-18;IGHJ4-1 gene-derived nAbs (hRSV12 and hRSV131) (Mousa *et al*, 2017) which have also been shown to compete against D25&Mota. Furthermore, the H-CDR3-inferred germline precursor sequence for IGHV1-18 mAbs shows three consecutive alanines which are flanked only on one side by an electrostatically charged residue (CARDTP-SIAAARLFDYW). These data suggest that V1-18 gene-derived nAbs undergo somatic hypermutation SHM and evolve toward this RSV preF-specific motif. Structural and/or mutagenesis studies would be useful to examine the role of each residue in binding to preF.

## Expanded IGHV1-derived clonotype families recognize an important vulnerability region on the preF surface

We selected and analyzed the most expanded clonotype families of nAbs to deeply investigate the humoral response against RSV

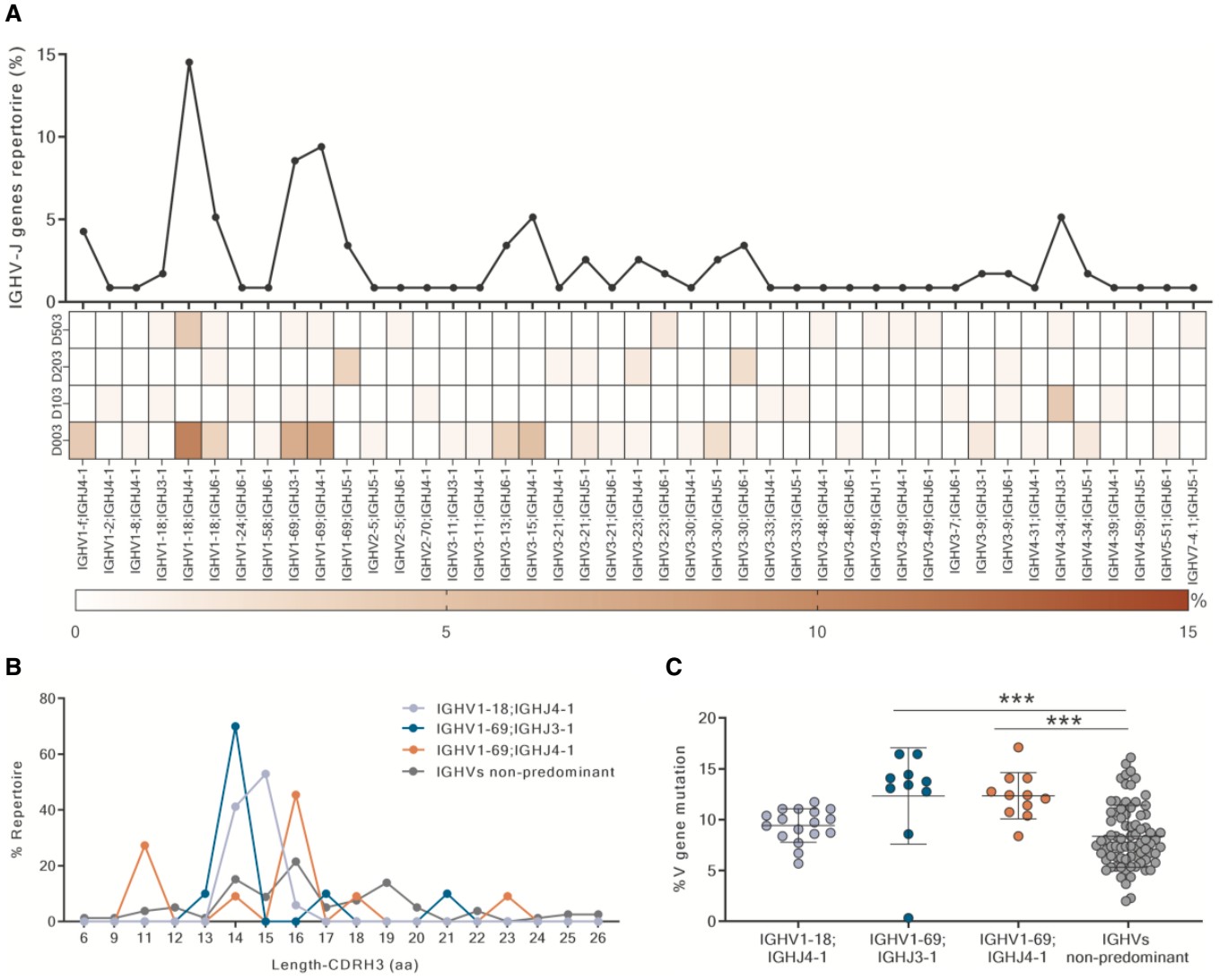

**Figure 3. Functional repertoire analysis of RSV F-protein-specific nAbs heavy and light chains.**

A  The graph shows the percentage of IGHV;IGHJ gene families observed in the functional repertoire and their distribution among the four donors. On the Y-axis is reported the percentage of heavy chain V-J gene rearrangements in the whole repertoire.

B  The diagram shows amino acid length of H-CDR3 region in predominant and non-predominant heavy chain rearrangements. The Y-axis reports the percentage of heavy chain gene usage within the specific V-J gene rearrangement repertoire.

C  The diagram shows the percentage of V-gene mutation frequencies in predominant and non-predominant heavy chain rearrangements. Mean and standard deviations are denoted on the graph. The P-value was calculated with the nonparametric One-way ANOVA test, and significance is shown as * ($P \leq 0.05$), ** ($P \leq 0.01$), *** ($P \leq 0.001$), and **** ($P \leq 0.0001$).

infection. To be grouped under the same clonotype family, antibodies need to share three main features: (i) Same V-J gene rearrangement; (ii) Same H-CDR3 sequence length; (iii) Highly similar H-CDR3 amino acid sequence (at least 75% nucleotide identity) (Kepler et al, 2014; Greiff et al, 2017). The IGHV1 gene family was found to be, in 3 out of 4 subjects, the most predominant among the expanded antibody clonotype families (Fig 5), confirming the predominance and expansion of this family in the whole functional repertoire.

nAbs belonging to the most expanded clonotype families (IGHV1-18 and IGHV1-69) presenting different levels of V-gene SHMs were selected for expression as recombinant full-length IgG to further characterize their binding and neutralization potency (Table 1). To better evaluate the preferential binding toward one of the two different F-protein conformations, all expressed nAbs were tested in the Gyrolab assay at the same concentration against both preF and postF, and the FI ratio was used as a measure to evaluate their binding preference (Fig 6A). Interestingly, all predominant gene-derived nAbs (IGHV1-18;IGHJ4-1 and IGHV1-69;IGHJ3-1) showed a significantly higher binding intensity toward the preF compared to non-predominant gene-derived nAbs. In several cases, the binding intensity was higher than the Site Ø-directed D25 (Fig 6A).

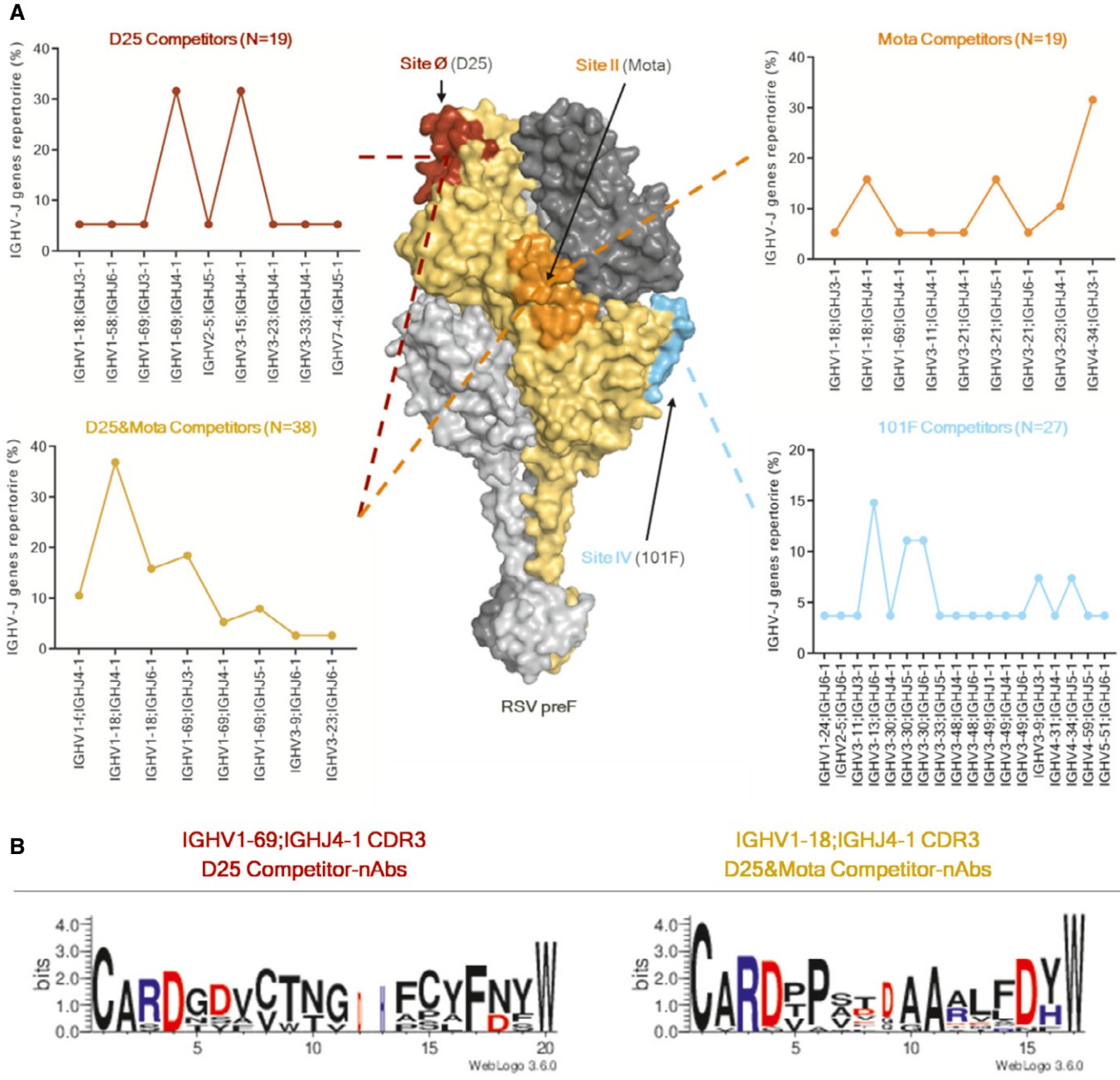

**Figure 4. Competitor nAbs heavy chain V–J gene rearrangement and H-CDR3s analysis.**

A  RSV preF trimer is shown as surface, and Site Ø (D25), Site II (Mota), and Site IV (101F) are highlighted in dark red, orange, and cyan, respectively. Each graph shows on the Y-axis the IGHV-J gene rearrangement % and on the X-axis all the nAbs heavy chain V-J gene rearrangements. All identified nAbs were pulled together and divided based on the specific epitope region for which they compete. The Y-axis reports the percentage of V-J gene rearrangements in the repertoire of competing nAbs.

B  Sequence logos representing the aligned amino acid sequences of nAbs grouped by the epitope region to which they bind. Only amino acids with electrostatically charged side chains have been colored. Blue and red letters are positively and negatively charged amino acids, respectively.

Following binding assessments, recombinant nAbs were tested for their neutralization potency *in vitro*. Intriguingly, all nAbs belonging to expanded clonotype families showed a very similar level of neutralization independent of their V-J gene usage and pre/postF binding preference. Indeed, the average half-maximal inhibitory concentrations (IC50) of clonotype expanded IGHV1-18; IGHJ4-1 and IGHV1-69;IGHJ3-1 gene-derived nAbs were 84.3 and 78.4 ng/ml respectively. These values were similar to the average

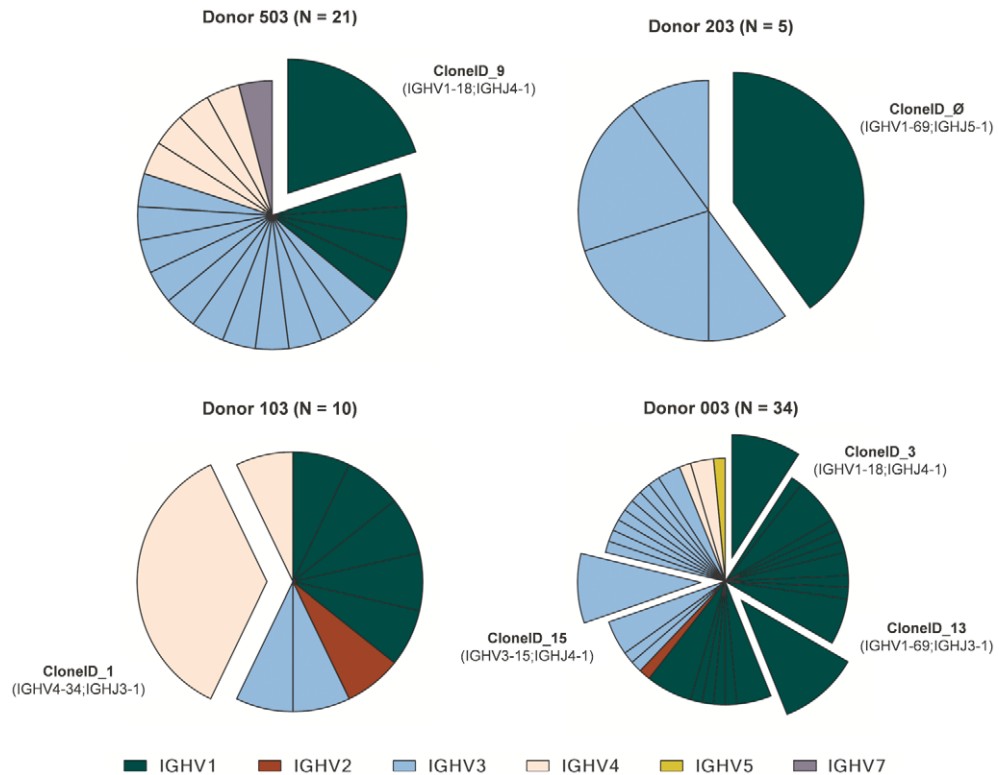

**Figure 5. Neutralizing antibody clonotype family analysis.**

Heavy chain sequences have been analyzed and grouped into clonotype families. The total number of families (N) identified per donor is shown in parenthesis. Expanded clonotype families are shown as exploded slices of the pie chart. Each color identifies a specific IGHV-gene family, where IGHV-1-2-3-4-5-7 are colored in green, red, light blue, pink, gold, and purple, respectively.

IC50 of non-predominant gene-derived nAbs (81.9 ng/ml) (Fig 6B). Interestingly, despite major binding fluorescence intensity toward the preF was observed for IGHV1-18;IGHJ4-1 and IGHV1-69;IGHJ3-1 gene-derived nAbs, these nAbs were found to be fivefold less neutralizing than the Site Ø-directed nAb D25 (Fig 6B).

Finally, we aimed to further characterize the epitope region recognized by predominant and non-predominant nAbs belonging to the most expanded clonotype families (Fig 6C). To do so, we implemented a competition assay using a more comprehensive panel of known nAbs that allowed us to cover all 6 neutralizing epitopes previously identified (Site Ø, Site I, Site II, Site III, Site IV, and Site V (Graham, 2019)). Only clonal families presenting at least four members were assessed. Fascinatingly, predominant gene-derived nAbs (IGHV1-18;IGHJ4-1 and IGHV1-69;IGHJ3-1) recognized a very similar epitope region. IGHV1-18;IGHJ4-1-derived nAbs, independently from the subject from which they were isolated, compete with D25 (Site Ø) (McLellan et al, 2013), Mota (Site II) (McLellan et al, 2010b), and MPE8 (Site III) (Corti et al, 2013). Some of them also competed with AM14, which recognizes a cross-protomer prefusion-specific epitope (Site V) (Gilman et al, 2015) (top and middle left bar charts). While IGHV1-69;IGHJ3-1-derived nAbs also compete with D25 (Site Ø) and Mota (Site II), all of them also compete with AM14 (Site V), and none compete with MPE8 (Site III) (bottom left bar chart). Therefore, the above described data suggest that these nAbs recognize a very similar

epitope region on the RSV preF surface despite showing a slightly different competition profile, and that they may recognize an important vulnerability region located between Site Ø (D25), Site II (Mota), and Site V (AM14). In contrast, non-predominant IGHV4-34;IGHJ3-1 gene-derived nAbs recognize a different region since they compete exclusively with Mota (Site II) and MPE8 (Site III) (bottom right bar chart).

# Discussion

Notwithstanding six decades of efforts, no vaccine against RSV is currently available and its heavy toll continues to impinge especially on older adults and children less than 5 years of age (Nair et al, 2010). Several vaccines against RSV are already under clinical evaluation (Mejias et al, 2020). Most of these vaccines target people already exposed to RSV infection such as pregnant women and older adults, and their antibody response to vaccination will build on the pre-existing immunity. Deep investigation of the repertoire of neutralizing antibodies is of high priority to support the development of vaccines against RSV.

The methodology applied in this study, integral to an approach named reverse vaccinology 2.0 (Rappuoli et al, 2016), allowed us to unravel the functional antibody repertoire of healthy adults and to identify 135 RSV F-protein-specific nAbs. Interestingly, we observed

**Table 1. Expanded clonotype family nAbs.**

| Donor ID | nAb ID | Clone ID | Heavy chain (V;J gene) | Light chain (V-J gene) | V-geneSHMs (%) | Binding specificity | Neutralization (IC$_{50}$ ng/ml) |
|---|---|---|---|---|---|---|---|
| D503 | PI1_E07 | 9 | 1-18;4-1 | 2-30;1-1 | 9.1 | preF | 57 |
| D503 | PI1_L06 | 9 | 1-18;4-1 | 2-30;1-1 | 9.7 | preF | 116 |
| D503 | PI1_G12 | 9 | 1-18;4-1 | 2-30;1-1 | 8.4 | preF | 114 |
| D203 | PI1_N12 | Ø | 1-69;5-1 | 3-11;3-1 | 12.4 | preF | 70 |
| D203 | PI1_L09 | Ø | 1-69;5-1 | 3-11;3-1 | 14.1 | preF | 62 |
| D103 | PI1_F02 | 1 | 4-34;3-1 | 1-39;2-1 | 6.4 | Cross-binder | 97 |
| D103 | PI1_L03 | 1 | 4-34;3-1 | 1-39;2-1 | 6.8 | Cross-binder | 63 |
| D103 | PI1_O20 | 1 | 4-34;3-1 | 1-39;2-1 | 7.1 | Cross-binder | 107 |
| D103 | PI1_I19 | 1 | 4-34;3-1 | 1-39;2-1 | 8.1 | Cross-binder | 54 |
| D003 | PI1_I19 | 13 | 1-69;3-1 | 3-20;1-1 | 13.4 | preF | 47 |
| D003 | PI1_B06 | 13 | 1-69;3-1 | 3-20;1-1 | 13.8 | preF | 61 |
| D003 | PI2_E08 | 13 | 1-69;3-1 | 3-20;1-1 | 14.4 | preF | 46 |
| D003 | PI2_C12 | 13 | 1-69;3-1 | 3-20;1-1 | 16.4 | preF | 160 |
| D003 | PI2_M03 | 3 | 1-18;4-1 | 2-30;2-1 | 8.6 | preF | 59 |
| D003 | PI2_I12 | 3 | 1-18;4-1 | 2-30;2-1 | 9.4 | preF | 94 |
| D003 | PI2_K18 | 3 | 1-18;4-1 | 2-30;2-1 | 10.1 | preF | 79 |
| D003 | PI2_M14 | 3 | 1-18;4-1 | 2-30;2-1 | 10.4 | preF | 61 |
| D003 | PI2_C04 | 3 | 1-18;4-1 | 2-30;2-1 | 10.8 | preF | 82 |
| D003 | PI3_B13 | 3 | 1-18;4-1 | 2-30;2-1 | 11.1 | preF | 96 |
| D003 | PI2_C13 | 15 | 3-15;4-1 | 1-9;2-1 | 7.2 | preF | 120 |

This table summarizing all expanded clonotype families nAb details following functional and genetic characterization.

that the majority of identified nAbs elicited following natural infection were able to bind both the preF and postF conformations (cross-binder nAbs) despite previous observations that nAbs able to recognize epitopes exclusive for the preF (Site Ø and V) are the most neutralizing (Graham, 2019). Therefore, RSV F-protein epitopes, across preF and postF, are well exposed on the viral surface, maintaining a certain degree of immunogenicity. This scenario is consistent with a previous electron cryotomography report showing that RSV naturally exposes on the mature viral surface high levels of postF particles (Liljeroos et al, 2013). While the preF-to-postF structural rearrangement is central to viral–host membrane fusion, it is possible to hypothesized that subsequent surface display of postF structures may have evolved in order to make highly available cross-conformational epitope regions to direct the antibody response toward less protective epitopes (Gilman et al, 2016; Graham, 2019). In this study, only one subject (Donor 003) presented a significantly higher level of preF-exclusive binding mAbs and consequent significantly higher percentage of nAbs. These data suggest that Donor 003 could have been more frequently or recently exposed to RSV infection leading to a higher level of protecting nAbs and further highlight the importance of preF targeting mAbs.

A major goal of this study was to unravel the RSV F-specific predominant gene rearrangements in the human functional antibody repertoire of four healthy adult donors. Public databases of either naïve or memory B-cell receptor (BCRs) repertoires have shown that the most used heavy chain V-gene families in humans are the IGHV3 and 4 (Briney et al, 2012; DeWitt et al, 2016; Soto et al, 2019), while

we have identified two key rearrangements belonging to the IGHV1 gene family (IGHV1-18 and IGHV1-69). A similar skew toward these two heavy chain V genes was also previously observed by Gilman et al (Gilman et al, 2016) showing consistency in the public repertoire of RSV F-specific antibodies. Following deep characterization of the heavy chain V-J gene usage of RSV F-specific nAbs, we identified three predominant rearrangements presenting the IGHV1-18;IGHJ4-1, IGHV1-69;IGHJ3-1, and IGHV1-69;IGHJ4-1 genes. Among these, IGHV1-18 seems to be more specific for the RSV F-protein repertoire since usage of IGHV1-69 has been observed for a variety of other pathogens, particularly influenza HA (Wu et al, 2015; Avnir et al, 2016; Avnir et al, 2017; Yassine et al, 2018; Sangesland et al, 2019) and HIV-1 (Bonsignori et al, 2016). It is worth noting that the average SHM level of RSV F-protein-specific IGHV1-69-derived nAbs (14.6%) is higher compared to predominant IGHV1-18-derived nAbs (9.4%) and to non-predominant gene-derived nAbs (8.4%). These data suggest that IGHV1-69 antibodies undergo more extensive maturation processes to acquire functional activity against the RSV F-protein despite showing an innate predisposition against viral glycoprotein targets. On the other hand, IGHV1-18-derived nAbs present a lower average SHM level suggesting that this lineage could better fit, in its germline conformation, the binding toward this antigen. Deeper investigation of IGHV1-18;IGHJ4-1 nAbs led to the identification of a conserved motif in H-CDR3 which was also seen in two previously identified RSV F-specific nAbs presenting the same heavy chain V-J gene rearrangement (Mousa et al, 2017) and it may constitute an important binding determinant.

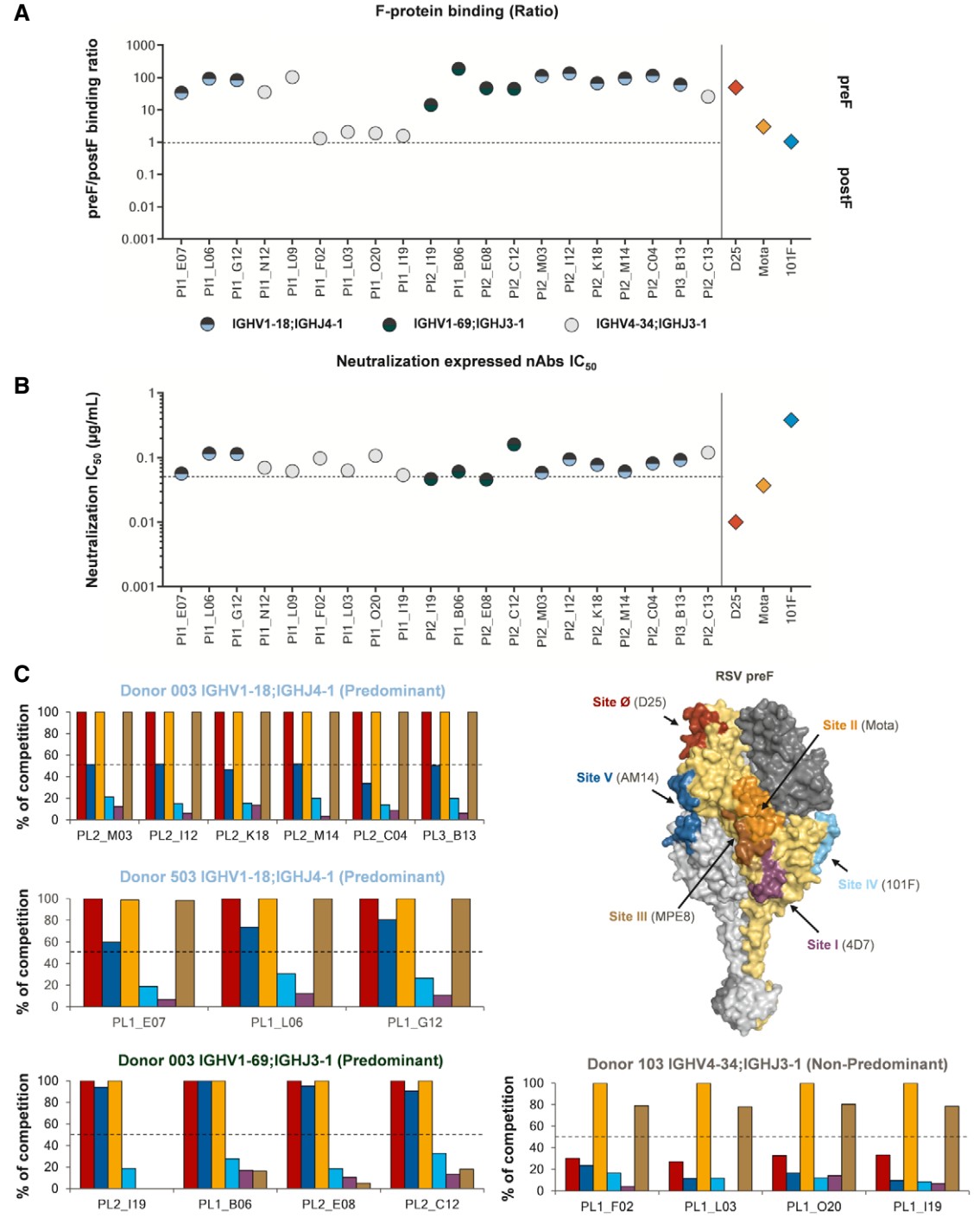

**Figure 6.  Deep functional characterization of clonotype expanded nAbs.**

A   nAbs were normalized for their concentration and assessed for their binding capability against the RSV F-protein. The graph shows on the Y-axis the ratio between preF and postF. nAbs above the threshold better recognize the F-protein in its preF conformation, while those under the threshold better recognize the postF state. Predominant gene-derived nAbs are shown in light blue and green dots for the IGHV1-18;IGHJ4-1 and IGHV1-69;IGHJ3-1, respectively. Non-predominant gene-derived nAbs are shown as gray dots.

B   nAbs were normalized and assessed for their neutralization potency, and the Y-axis shows the IC$_{50}$ in µg/ml. Each value is an average of three technical replicates. Predominant gene-derived nAbs are shown in light blue and green dots for the IGHV1-18;IGHJ4-1 and IGHV1-69;IGHJ3-1, respectively. Non-predominant gene-derived nAbs are shown as gray dots.

C   Deep competition assay of expanded clonotype families is represented as a graph bar, where the Y-axis shows the % of competition with known neutralizing nAbs. Each bar is colored as the highlighted epitope shown on the surface of the RSV preF trimer.

The functional repertoire analyses reported here support two main immunological concepts. The first one is that naïve B-cell germlines are specifically selected for expansion and immunodominance based on their functionality when they first encounter the cognate antigen. The second concept is that selected antibody germlines may have evolved to better recognize, in their naïve conformations, specific protein and structural "patterns" as was also demonstrated for IGHV1-69-derived bnAbs against HA-stem of influenza (Lingwood *et al*, 2012; Avnir *et al*, 2016). To investigate the epitope regions on the preF surface that drive the expansion of nAbs following natural infection, we analyzed the most expanded clonotype families among donors. Interestingly, while the expanded clonotype family presenting the non-predominant gene rearrangement IGHV4-34;IGHJ3-1 recognizes a portion in the middle of RSV preF (Site II and Site III), both predominant gene-derived IGHV1-18;IGHJ4-1 and IGHV1-69;IGHJ3-1 nAbs were able to recognize a restricted region exposed exclusively on top of the RSV preF globular head which is placed among Site Ø, II/III, and V. These data highlight differences between the adults and infants antibody repertoire where the most predominant heavy chain V-gene family was seen to be the IGHV3 (IGHV3-21;IGLV1-40) and the majority of naïve antibodies were directed exclusively against a broader yet less neutralizing epitope region which is shared between the preF and postF surfaces (Site III) (Corti *et al*, 2013; Goodwin *et al*, 2018). This scenario suggests that multiple exposures to RSV could be necessary to mature the population of predominant gene-derived nAbs observed in adults (IGHV1-18; IGHJ4-1–IGHV1-69;IGHJ3-1/4-1).

Based on all the observations mentioned above, the development of RSV vaccines based on the preF antigen could be of extreme value. From an immunological point of view, immunization with the stabilized preF trimmer could boost the maturation of cross-binding nAbs (the most abundant class of nAbs elicited following natural infection) toward more immunogenic regions, as well as lead to further expansion of predominant gene-derived preF-specific nAbs. This hypothesis is also supported by a recent study, aimed to evaluate the effectiveness of a structurally designed preF antigen (DS-Cav1), which showed that the majority of F-protein binding antibodies were directed toward shared surfaces on both preF and postF, and sera absorption of preF-specific antibodies abrogated almost completely RSV neutralization (Crank *et al*, 2019). From a vaccine strategy point of view, immunization with a preF antigen could benefit at-risk populations (infants < 6 months of age and frail elderly) as well as not at-risk populations (pregnant women and healthy adults). In fact, maternal immunization with a preF antigen could lead to passive protection of infants by a robust preF-directed polyclonal response, while immunization of healthy adults could elicit a more robust immunological memory and, since older adults mainly use MBCs to respond against infection or vaccination, to a more efficient antibody response against RSV in advanced age. Overall, this work defines the functional and genetic characteristics of the preF antibody functional repertoire elicited following RSV natural infection in healthy adults and provides novel insights, at single-cell level, on how vaccination with a preF immunogen could drive the pre-existing immunity and power the functional antibody response leading to protection against RSV infection.

# Materials and Methods

## Human samples

This work was supported by a collaboration with Azienda Ospedaliera Empoli (IT) that provided samples from healthy blood donors who previously gave their written consent. The study was approved by local ethics committees and conducted according to good clinical practice in accordance with the declaration of Helsinki (European Council 2001, US Code of Federal Regulations, ICH 1997). The experiments performed with samples derived from this study conformed to the principles set out in the Department of Health and Human Services Belmont Report. This study was unblinded and not randomized.

## Single cell sorting and culture of RSV preF$^+$ memory B cells

Human peripheral blood mononuclear cells (hPBMCs) from adult healthy donors were pre-enriched for the B-cell population using the EasySep enrichment kit (Stemcell technologies) following the manufacturer's protocol. Cells were stained with Live/Dead Fixable Aqua (Invitrogen; Thermo Scientific) in 100 μl final volume diluted 1:500 at room temperature (RT) for 20 min and washed twice with phosphate-buffered saline (PBS). Unspecific bindings were saturated with 50 μl of rabbit serum 20% in PBS and incubated at 4°C for 20 min and washed twice with PBS. Cells were then stained with CD19 V450 (BD cat# 560353), IgM-FITC (BD cat# 555782), IgA-FITC (Jackson cat# 109-096-011), and IgD-A700 (BD cat# 561302) diluted 1:25, 1:20, 1:40, and 1:15 respectively together with the antigen RSV preF-Alexa647 used at 0.3 μg/ml in 1% FBS at 4°C for 1 h. Stained MBCs were single cell-sorted with a BD FACSAria III (BD Biosciences) into 384-well plates containing 3T3-CD40L feeder cells and were incubated with IL-2 and IL-21 for 14 days as previously described (Huang *et al*, 2013).

## Human IgG screening by ELISA

Enzyme-linked immunosorbent assay (ELISA) was used to detect the presence of human IgG within the supernatant of cultured single cell-sorted preF$^+$ MBCs. 384-well plates (Nunc MaxiSorp 384-well plates; Sigma-Aldrich) were coated with 2 μg/ml unconjugated goat anti-human IgG (Sigma-Aldrich) and incubated at 4°C overnight. Plates were then washed three times with PBS/tween 0.05%, and 50 μl/well of blocking solution (PBS/BSA 1%) was used to saturate unspecific binding. Plates were incubated at 37°C for 1 h without CO$_2$ and then washed three times with PBS/tween 0.05%. Supernatants were diluted 1:2.5 in PBS/BSA 0.1%/Tween20 0.05% in 25 μl/well final volume and incubated for 2 h at 37°C without CO$_2$. 25 μl/well of alkaline phosphatase-conjugated goat anti-human IgG (Sigma-Aldrich) was used as secondary antibody and, following three washes with PBS/tween 0.05%, PNPP (p-nitrophenyl phosphate) (ThermoFisher) was used as soluble substrate to detect the presence of human IgG. The final reaction was measured in a spectrophotometer at a wavelength of 450 nm.

## Gyrolab immunoassay for screening of IgG antigen-specific binding

Antibody binding toward the RSV F-protein, in either preF or postF conformation, was performed by Gyrolab microfluidic immunoassays

as previously described (Giuliani *et al*, 2018). All supernatants containing naturally produced IgGs were diluted 1:2 in Rexxip H-Max (Gyros) and analyzed using Gyrolab Bioaffy 200 CDs (Gyros) and the standard Gyrolab three-step method (capture–analyte–detection) for qualitative screening. Biotinylated preF and postF (captures) were prepared using the EZ-Link sulfo-NHS-LC-Biotin (Thermo Scientific) at a molar ratio of 10 biotin : 1 protein and used at 100 μg/ml (diluted in PBS/Tween20 0.01%). Alexa 647-conjugated Goat Anti-Human IgG (Jackson Immuresearch) was used as detection reagent and prepared at 25 nM diluted in Rexipp F buffer (Gyros). To quantify IgG within each supernatant, a standard curve was prepared by using purified human IgG (Sigma-Aldrich) starting at 4 μg/ml and diluted step 1:4 in Rexxip H (Gyros), while samples were diluted 1:4 in Rexxip H (Gyros). Following their expression, mAbs were further screened for quantitative binding to both preF and postF, and capture/detection reagents were prepared as described above. Samples and known neutralizing antibodies (D25, Mota and 101F) were prepared at a starting concentration of 1 μg/ml in dose–response curves to select the range of linearity of the response.

## Plaque reduction neutralization assays for mAbs functional screening

96-well TC-Treated Microplates (Greiner Bio-One) were coated with 70–80% confluent Vero cells (ATCC) at 320.000 cells/ml (100 μl/well) and incubated for 3 h at 37°C 5% $CO_2$. For high-throughput and qualitative screening, mAbs were diluted at a single point (1:5) in 25 μl/well of PBS/FBS 5% and co-incubated at 1:1 (mAb:virus) volume ratio for 1h at 37°C 5% $CO_2$ with live RSV long-strain (MOI 0.8) diluted 1:110 in PBS/FBS 5%. Virus dilution was selected based on the ability to produce around 120 syncytia/well on Vero cells when tested alone. Known neutralizing mAbs used as positive control (D25, Mota, and 101F) were used at 5 μg/ml and 500 ng/ml. Following incubation, 25 μl/well of the above described mAb/virus mix was placed on the Vero cells and incubated for 2 h at 37°C 5% $CO_2$. The mix was then replaced with 100 μl/well of infection medium (1.5% methylcellulose + 2× EMEM/FBS 10%/100× Glutamax/100× penicillin–streptomycin [v:v = 1:1]) and cells incubated at 37°C 5% $CO_2$ for 72 h. On the third day, infection medium was removed and cells were fixed with 100 μl/well of 4% formalin solution (Sigma-Aldrich). After 1 h incubation, cells were washed three times with 250 μl/well of PBS and surfaces permeabilized with 100 μl/well of permeabilization buffer (PBS/saponin 0.5% (Sigma-Aldrich)/FBS 2.5%) incubated for 1 h at RT. Cells were washed three times with PBS/Tween20 0.1% (Sigma-Aldrich) and 100 μl/well of PBS/BSA 1% was added for 1 h at RT. Cells were washed three times with PBS/Tween20 0.1% (Sigma-Aldrich), and 100 μl/well of mouse α-RSV F IgG2b/mouse α-RSV nucleoprotein IgG1 mix (Bio-Rad) diluted 1:1,000 in permeabilization buffer was added for 1 h at RT. Cells were washed three times with PBS/Tween20 0.1% (Sigma-Aldrich), and 100 μl/well of Goat F(ab')2 α-mouse IgG (H + L) Human ads-HRP (SouthernBiotech) diluted 1:4,000 in permeabilization buffer was added for 1 h at RT. Cells were washed three times with PBS/Tween20 0.1% (Sigma-Aldrich), and 100 μl/well of KPL TrueBlue Peroxidase Substrate (Seracare) was added for 10 min at RT. Finally, cells were washed six times with dH₂O, dried for 2 h, and syncytia counted with an ImmunoSpotAnalyzer (CTL Europe). MAbs were considered neutralizing if capable to reduce the number of syncytia formation by at least 50% compared to the virus tested alone. For quantitative screening, recombinant nAbs and positive controls were tested at 3 μg/ml. Results were fit to sigmoidal dose–response curves with nonlinear regression, and $IC_{50}$ values were calculated using GraphPad Prism 7.

## Competition assay by Gyrolab workstation

Competition assay was performed as for the antigen-binding specificity method described above. Differences lay on sample preparation before binding assessment toward the preF antigen. Indeed, mAbs were mixed with one of three known neutralizing antibodies (D25, Mota, and 101F) at the time used at 100 μg/ml (which gave complete signal saturation when tested alone). The three known neutralizing antibodies were used in their Fab form and an Alexa 647-conjugated Goat Anti-Human IgG Fc-specific (Jackson Immuresearch) was used as detection reagent and prepared at 25 nM diluted in Rexipp F buffer. Competition was considered when preF-binding signal was reduced by at least 50% compared to the mAb tested alone.

## mAbs variable region RT–PCR, PCR, and sequencing

MBCs supernatant was previously recovered and lysed in 20 μl/well of lysis buffer (H₂O DEPC (Thermo Scientific)/0.2 U/μl RNaseOUT (Thermo Scientific)/ 1 mg/ml BSA (Thermo Scientific)). The reagent mix per well (20 μl) for single cell reverse transcription polymerase chain reaction (RT–PCR) of mAb variable heavy and light chain regions was composed by 1 μl dNTPs [10 mM] (Thermo Scientific), 2 μl DTT [0.1 M] (Thermo Scientific), 1 μl RNaseOUT [40 U/μl] (Thermo Scientific), 4 μl 5× First-Strand Buffer (Thermo Scientific), 5 μl MgCl₂ [25 mM] (Thermo Scientific), 0.25 μl Super-Script IV [200 U/μl] (Thermo Scientific), 0.37 μl of DPEC H20 (Thermo Scientific) 0.46 μl of each VH (IgG: GGAAGGTGTG-CACGCCGCTGGTC; IgA: CCTGGGGGAAGAAGCCCTGGACC), Vκ (CCTCTAACACTCTCCCCTGTTGAAG), and Vλ (CATTCTGYAGGG GCMACTGTCTTCTC) primer [10 μM] and 5 μl of cellular lysate. The RT–PCR was programmed for 60 min 50°C, 5 min 70°C, and 10°C on hold. Recovered cDNAs were used for two runs of polymerase chain reaction (PCR). PCRI was programmed starting with 98°C 30 s, followed by five cycles of 10 s 98°C, 20 s 57°C, 30 s 72°C, 30 cycles of 10 s 98°C, 20 s 60°C, 30 s 72°C, and 10°C on hold. The PCRI master mix per well (25 μl) was composed by 4 μl cDNA, 1.25 μl Fw primer mix [10 μM], 1.25 μl Rv primer mix [10 μM], 0.5 μl dNTPs [10 mM] (Thermo Scientific), 7.75 μl ddH₂O, 0.25 μl Q5 High-Fidelity DNA Polymerase (New England Biolabs), 5 μl 5× Q5 Buffer, and 5 μl 5× Q5 GC Enhancer. PCRII was programmed as for PCRI, while the master mix per well (25 μl) was composed by 3 μl PCRI-product, 1.25 μl Fw nested primer mix [10 μM], 1.25 μl Rv nested primer mix [10 μM], 0.5 μl dNTPs [10 mM] (Thermo Scientific), 8.75 μl ddH₂O, 0.25 μl Q5 High-Fidelity DNA Polymerase (New England Biolabs), 5 μl 5× Q5 Buffer, and 5 μl 5× Q5 GC Enhancer. PCRII products were stained with 6× DNA gel loading dye (Thermo Scientific) and loaded in 2% agarose gel electrophoresis. Recovered VH and VL PCRII products were purified using magnetic beads D-pure (Nimagen) or MagiSi-DT Removal (MagnaMedics) and Sanger-sequenced with an ABI3730xl DNA Analyzer (Applied Biosystems) using PCRII primer mix as described

above. Sequence reads were analyzed with Sequencher (Gene Codes Corporation), saved in FASTA format and the repertoire was explored using Cloanalyst (http://www.bu.edu/computationalimmu nology/research/software/) (Kepler, 2013; Kepler *et al*, 2014). Nonfunctional sequences, or those in which we were unable to identify the CDR3, were removed from the data set.

### Recombinant full-length human IgG expression and purification

Neutralizing antibodies in their full-length IgG form were expressed as previously described (Giuliani *et al,* 2018; Lopez-Sagaseta *et al,* 2018). Briefly, full-length IgG expression started with the ligation of 3 μl of selected VH and VL DNA into human pRS5a Igγ1, Igκ, and Igλ expression vectors (Novartis-NIBR), which were used to transform Mach1 competent *E. coli* (Thermo Scientific). Positively transformed colonies were expanded, and plasmid DNA extraction and purification were performed with E-Z 96 FastFilter Plasmid DNA Kit (Omega Bio-tek). Paired purified Ig heavy and Ig light chain DNA plasmids (15 μg each/30 ml of transfection volume) were used for transfection and transient expression in Expi293 cells (Thermo Fisher Scientific) which was performed according to manufacturer's protocol. Cell supernatant was centrifuged at 900× *g* for 10 min and filtered through 0.22 μm pore size filter (Millipore) to remove cell debris. Expressed mAbs were purified with Protein G Sepharose 4 Fast Flow (GE Healthcare) and eluted with 0.1 M glycine (pH 3.0) in tubes containing 1 M Tris (pH 9.0). Following purification and buffer exchange into PBS pH 7.4 through PD-10 Desalting Column (GE Healthcare), expressed recombinant mAbs were quantified by NanoDrop spectrophotometer (Thermo Scientific) (absorbance 280 nm) and purity assessed by SDS–PAGE on a 4–12% Bis-Tris Gel and Problue Safe Stain (Giotto Biotech).

### Biolayer interferometry for deep competition assay

Antibody competition was measured by the ability of known anti–RSV F mAbs (primary antibody) to inhibit binding of our expressed recombinant nAbs (secondary antibody) against RSV preF. The assay was performed by Octet Red384 (ForteBio Pall Life Sciences), a biolayer interferometry platform. Fc-specific Anti-Human IgG biosensor tips (ForteBio Pall Life Sciences) were equilibrated in 200 μl of PBS/BSA 1% prior use. Antibodies and the preF antigen were prepared at 20 and 4 μg/ml, respectively, and all reagents were diluted in PBS/BSA 1% into 384-well plates. The assay used a three-step method: (i) primary antibody loading onto biosensor tips; (ii) unconjugated antigen capturing to primary antibody; (iii) secondary antibody binding to primary antibody/antigen complex. Each association step (300s) was followed by a washing step (30s). Competition of primary antibody was observed when secondary antibody binding signal was inhibited by at least 50%.

### Statistical analysis

Since the study aimed to analyze the antibody functional repertoire of healthy adults, donors were selected, once they gave their written consent, exclusively on their willingness to participate to this study. Therefore, the study was unblinded and not randomized. Sample size was selected based on the availability of blood samples for this

**The paper explained**

**Problem**

Respiratory syncytial virus (RSV) is a leading cause of death from lower respiratory tract infection in high-risk populations, and all people are recurrently infected throughout their life. Today, several vaccines are under clinical development and these vaccines will stimulate the pre-existing antibody response. Therefore, it is pivotal to profile the protective antibody response elicited by RSV natural infection to understand and characterize the impact of vaccination in immunized individuals.

**Results**

We described an in-depth analyses of the repertoire of neutralizing antibodies (nAbs) following RSV natural infection. We observed that the majority of induced antibodies bind a region shared between the prefusion (preF) and postfusion (postF) conformations of the F-protein. A smaller fraction of nAbs are specific for the preF form and belong to the IGHV1 gene family. Intriguingly IGHV1-gene-derived nAbs are conserved among individuals and target an important site of pathogen vulnerability exclusive for the preF conformation.

**Impact**

To our knowledge, this is the first time that the human repertoire of nAbs against RSV is characterized in such depth. Our data are extremely relevant to understand the immune response induced by RSV vaccines since their response and effectiveness will build on their ability to engage and evolve the pre-existing memory B cells that we described in this manuscript. This information can accelerate the development of RSV vaccines to make them globally available in the shortest possible timeframe.

study. Statistical analysis was assessed with GraphPad Prism Version 7.00 (GraphPad Software, Inc., San Diego, CA). Nonparametric One-way ANOVA test was used to evaluate statistical significance of F-protein-specific and neutralizing antibodies among the four donors analyzed in this study. Statistical significance was shown as * for values $\leq 0.05$, ** for values $\leq 0.01$, *** for values $\leq 0.001$, and **** for values $\leq 0.0001$.

## Data availability

Nucleotide and amino acidic sequences of RSV-neutralizing antibodies identified in this study are available at GenBank, Pop-set name Homo sapiens anti-RSV immunoglobulin variable region mRNA, partial cds, accession number MW679370 - MW679409 (https://www.ncbi.nlm.nih.gov/popset?DbFrom=nuccore&Cmd=Link&LinkName=nuccore_popset&IdsFromResult=2017756031).

**Expanded View** for this article is available online.

### Acknowledgements

We wish to thank the Azienda Ospedaliera Empoli (IT) that provided samples from healthy blood donors under a study approved by local ethic committees and conducted according to good clinical practice in accordance with the declaration of Helsinki. Patients have given their written consent to the study. We wish to thank study participants. We also wish to thank Dr. Mark Connors for providing the following reagent obtained through the NIH AIDS Reagent Program, Division of AIDS, NIAID, NIH: Cat#12535 3T3-msCD40L cells.

## Author contributions

Conceiving this project: OF and FB. Experiments design: EA and FB. Experimental protocols: EA, IP, MBa, ST, CS, EF, SG, GT, and MBi. Experimental results analysis and writing the manuscript: EA. Intellectual contribution to finalize the manuscript: EA, UD, SC, MJB, RR, OF, and FB. All authors reviewed the manuscript.

## Conflict of interest

This work was sponsored by Novartis Vaccines Srl, now acquired by the GSK group of companies. The sponsor was involved in all stages of the study conducted and analysis. EA has participated in a postgraduate studentship program at GSK. IP was an employee of GSK group of companies. MB, ST, CS, EF, SG, GT, MB, UD, SC, MJB, RR, OF, and FB are employees of GSK group of companies.

## For more information

Author's website:
- https://it.gsk.com/it-it/chi-siamo/le-nostre-sedi-in-italia/gsk-vaccines-srl/

Relevant database:
- http://www.imgt.org/
- https://www.ncbi.nlm.nih.gov/genbank/

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
