## [Review Process File · EMBO Molecular Medicine]

The respiratory syncytial virus (RSV) prefusion F-protein functional antibody repertoire in adult healthy donors

Emanuele Andreano, Ida Paciello, Monia Bardelli, Simona Tavarini, Chiara Sammiceli, Elisabetta Frigimelica, Silvia Guidotti, Giulia Torricelli, Marco Biancucci, Ugo D'Oro, Sumana Chandramouli, Matthew Bottomley, Rino Rappuoli, Oretta Finco, and Francesca Buricchi

DOI: 10.15252/emmm.202114035

Corresponding author: Francesca Buricchi (francesca.x.buricchi@gsk.com)

Review Timeline:

Submission Date:	27th Jan 21
Editorial Decision:	12th Feb 21
Revision Received:	12th Mar 21
Editorial Decision:	17th Mar 21
Revision Received:	10th Apr 21
Accepted:	13th Apr 21

Editor: Zeljko Durdevic

Transaction Report:

12th Feb 2021

Dear Dr. Buricchi,

Thank you for the submission of your manuscript to EMBO Molecular Medicine. We have now received feedback from the two reviewers who agreed to evaluate your manuscript. As you will see from the reports below, the referees acknowledge the interest of the study and are overall supporting publication of your work pending appropriate revisions.

Addressing the reviewers' concerns in full will be necessary for further considering the manuscript in our journal, and acceptance of the manuscript might entail a second round of review. EMBO Molecular Medicine encourages a single round of revision only and therefore, acceptance or rejection of the manuscript will depend on the completeness of your responses included in the next, final version of the manuscript. For this reason, and to save you from any frustrations in the end, I would strongly advise against returning an incomplete revision.

I look forward to reading a new revised version of your manuscript as soon as possible.

Yours sincerely,

Zeljko Durdevic

***** Reviewer's comments *****

Referee #1 (Comments on Novelty/Model System for Author):

The authors do an excellent job examining the B cells and IgG neutralizing antibodies against RSV for 4 healthy individuals. When individuals are vaccinated with an RSV prefusion F protein in current clinical trials of pregnant women to boost their transplacental antibodies to their fetus or of elders to protect them from lethal RSV infection, these are the B cells and antibodies most likely to be induced and to neutralize RSV upon infection: infant or elder. This work will make it easier to characterize the response in these immunized individuals. Analysis of more healthy individuals would strengthen the study, but that will likely come later.

Referee #1 (Remarks for Author):

Andreano et al. have isolated B cells from 4 healthy individuals and identified B cells producing neutralizing IgG type antibodies to RSV and characterized the binding activities of these nAbs that recognize preF vs postF (the active and the refolded, non-functional) forms of the F protein. Pregnant women and elders are currently being immunized with preF in clinical trials, and if these trials are successful, many more individuals will be immunized in the future. The preF immunogen will likely expand the existing B cell response of these cells. They investigators found that most nAbs bound to both the preF and postF forms of the F protein (cross-binders) while a smaller group bound exclusively to preF. Cross-binder nAbs used a wide array of gene rearrangements, while

preF-binder nAbs derived mostly from the expansion of B cell from IGHV1 germline clonotypes which recognize an epitope between Site Ø, Site II and Site V on the F-protein, identifying an important neutralization site.

Of the 4 donors they followed, one produced only cross-binding neutralizing nAbs and another produced mainly preF specific nAbs. The other two produced predominantly cross-binders.

Although the investigators isolated and examined B cells from only 4 individuals, they were able to define nAbs that inhibited known mAbs binding neighboring antigenic sites unique to preF and others that are common to both preF and postF, characterized them as cross-binders. This central antigenic site appears to be a common neutralization site, adding to those already described.

Minor suggestions:

l.31. death, not plural.

l.51. no need to use "around" when providing a range.

Fig. 2. What dilution of antibody was used to determine the neutralization activity of each clone?

Fig. 6A and B. It would be helpful to label the Y axes. Which region indicates preFpostF.

Referee #2 (Remarks for Author):

The authors have effectively shown that eliciting antibodies to the pre-fusion F protein is needed to developing an RSV vaccine. This manuscript provides a large amount of data from human donors that gives insight into the natural human antibody diversity to the RSV F protein and the importance of specific 'cross-binding' antibodies that bind to both the pre and post-fusion F protein conformations. The manuscript is well-written with only a few minor revisions.

- Introduction- Consider outlining the epitopes mapped to F protein - highlight the different epitopes present on pre- vs post-F protein
- Methods- Please clarify line 421 - "diluted step 4" - what is step 4?
- -Figure 1A: Please put the appropriate amount of stars to indicate significance between donor 103 and donor 003.
- -Figure 2: Adding a pymol or cartoon on the F-protein with the antigenic sites would greatly enhance the clarity of this figure
- Figure 3C label is cut off and lines for IGHVs non-predominant are covered by dots; please clarify if ranges shown are SEM/C/SD
- Please add Y-axis labels to Figure 6A/B

Answers to referees comments**General comment**

We thank the referees for their critical assessment of our work. In the revised version of the manuscript we have modified the text and figures in accordance to Referee comments and suggestions.

Furthermore, as required by the revision process, all figures were saved in CMYK 300 DPI, supplemental figures were renamed Expanded View (EV), the checklist was completed, corresponding author ORCID ID was provided, a "Data availability" session was added including the GenBank accession number for deposited antibody sequences, and summary and synopsis were written and submitted.

All changes in the text were highlighted in yellow to be clearly visible.

For clarity purposes, the authors also removed colors from Table 1 and Fig. 6A – B Y axis were labeled.

The authors also prepared a graphical abstract for this manuscript.

Referee comments**Referee #1 (Comments on Novelty/Model System for Author):**

The authors do an excellent job examining the B cells and IgG neutralizing antibodies against RSV for 4 healthy individuals. When individuals are vaccinated with an RSV prefusion F protein in current clinical trials of pregnant women to boost their transplacental antibodies to their fetus or of elders to protect them from lethal RSV infection, these are the B cells and antibodies most likely to be induced and to neutralize RSV upon infection: infant or elder. This work will make it easier to characterize the response in these immunized individuals. Analysis of more healthy individuals would strengthen the study, but that will likely come later.

Referee #1 (Remarks for Author):

Andreano et al. have isolated B cells from 4 healthy individuals and identified B cells producing neutralizing IgG type antibodies to RSV and characterized the binding activities of these nAbs that recognize preF vs postF (the active and the refolded, non-functional) forms of the F protein. Pregnant women and elders are currently being immunized with preF in clinical trials, and if these trials are successful, many more individuals will be immunized in the future. The preF immunogen will likely expand the existing B cell response of these cells. They investigators found that most nAbs bound to both the preF and postF forms of the F protein (cross-binders) while a smaller group bound exclusively to preF. Cross-binder nAbs used a wide array of gene rearrangements, while preF-binder nAbs derived mostly from the expansion of B cell from IGHV1 germline clonotypes which recognize an epitope between Site Ø, Site II and Site V on the F-protein, identifying an important neutralization site.

Of the 4 donors they followed, one produced only cross-binding neutralizing nAbs and another produced mainly preF specific nAbs. The other two produced predominantly cross-binders.

Although the investigators isolated and examined B cells from only 4 individuals, they were able to define nAbs that inhibited known mAbs binding neighboring antigenic sites unique to preF and others that are common to both preF and postF, characterized them as cross-binders. This central antigenic site appears to be a common neutralization site, adding to those already described.

Minor suggestions:

1.31. death, not plural.

1.51. no need to use "around" when providing a range.

Fig. 2. What dilution of antibody was used to determine the neutralization activity of each clone?

Fig. 6A and B. It would be helpful to label the Y axes. Which region indicates preFpostF.

Referee #2 (Remarks for Author):

The authors have effectively shown that eliciting antibodies to the pre-fusion F protein is needed to developing an RSV vaccine. This manuscript provides a large amount of data from human donors that gives insight into the natural human antibody diversity to the RSV F protein and the importance of specific 'cross-binding' antibodies that bind to both the pre and post-fusion F protein conformations. The manuscript is well-written with only a few minor revisions.

- Introduction- Consider outlining the epitopes mapped to F protein - highlight the different epitopes present on pre- vs post-F protein
- Methods- Please clarify line 421 - "diluted step 4" - what is step 4?
- Figure 1A: Please put the appropriate amount of stars to indicate significance between donor 103 and donor 003.
- Figure 2: Adding a pymol or cartoon on the F-protein with the antigenic sites would greatly enhance the clarity of this figure
- Figure 3C label is cut off and lines for IGHVs non-predominant are covered by dots; please clarify if ranges shown are SEM/CI/SD
- Please add Y-axis labels to Figure 6A/B

Point by point response to the Referees**Referee 1**

Referee 1 P1: 1.31. death, not plural.

Answer P1: We modified the text accordingly.

Referee 1 P2: 1.51. no need to use "around" when providing a range.

Answer P2: We deleted the word "around" from the text.

Referee 1 P3: Fig. 2. What dilution of antibody was used to determine the neutralization activity of each clone?

Answer P3: We added in Fig. 2 legend (line 815 – 816) the dilution used to screen our monoclonal antibodies which is 1:5. The same information is reported in the material and methods section at line 450 - 451.

Referee 1 P4: Fig. 6A and B. It would be helpful to label the Y axes. Which region indicates preFpostF.

Answer P4: We labeled the Y axes of Fig. 6A and B.

Referee 2

Referee 2 P1: Introduction- Consider outlining the epitopes mapped to F protein - highlight the different epitopes present on pre- vs post-F protein

Answer P1: We have outlined in the text where the different immunogenic epitopes on the F-protein are (line 77 – 80).

Referee 2 P2: Methods- Please clarify line 421 - "diluted step 4" - what is step 4?

Answer P2: We modified the text and wrote "diluted 1:4" instead of "step 4"; now line 441.

Referee 2 P3: Figure 1A: Please put the appropriate amount of stars to indicate significance between donor 103 and donor 003.

Answer P3: The appropriate amount of stars were indicated in Figure 1A as suggested by the Referee. In addition, for consistency, the authors also added the appropriate amount of stars in Figure 2A.

Referee 2 P4: Figure 2: Adding a pymol or cartoon on the F-protein with the antigenic sites would greatly enhance the clarity of this figure

Answer P4: The 3D structure of the F-protein highlighting the immunogenic sites described in the figure were added as suggested by the Referee.

Referee 2 P5: Figure 3C label is cut off and lines for IGHVs non-predominant are covered by dots; please clarify if ranges shown are SEM/CI/SD

Answer P5: We have added the label to figure 3C, and ranges were clarified in the figure legend at line 829 – 830.

Referee 2 P6: Please add Y-axis labels to Figure 6A/B

Answer P6: We labeled the Y axes of Fig. 6A and B as already stated in “Answer P4” to Referee 1

17th Mar 2021

Dear Dr. Buricchi,

Thank you for the submission of your revised manuscript to EMBO Molecular Medicine. I am pleased to inform you that we will be able to accept your manuscript pending the following final amendments:

1) In the main manuscript file, please do the following:

- Add up to 5 keywords.
- Remove text highlight color.
- Make sure that all special characters display well.
- Specify in author contribution MB for Monia Bardelli and Marco Biancucci (e.g MoBa and MaBi).
- In M&M, provide primer sequences used for the RT-PCR reactions.
- In M&M, include that, in addition to the WMA Declaration of Helsinki, the experiments conformed to the principles set out in the Department of Health and Human Services Belmont Report.
- In M&M, statistical paragraph should reflect all information that you have filled in the Authors Checklist, especially regarding randomization, blinding, replication etc.
- First reference is duplicated, please remove one. In the reference list, citations should be listed in alphabetical order. Where there are more than 10 authors on a paper, 10 will be listed, followed by "et al.". Please check "Author Guidelines" for more information.

<https://www.embopress.org/page/journal/17574684/authorguide#referencesformat>

- In addition to the accession number please provide URL for nucleotide and amino acid sequences of RSV neutralizing antibodies. Please be aware that all datasets should be made freely available upon acceptance, without restriction. Use the following format to report the accession number of your data:

[data type]: [full name of the resource] [accession number/identifier] ([doi or URL or identifiers.org/DATABASE:ACCESSION])

Please check "Author Guidelines" for more information.

<https://www.embopress.org/page/journal/17574684/authorguide#availabilityofpublishedmaterial>

2) Author Checklist: Please fill in fields 6 and 7.

3) The Paper Explained: Please add it to the main manuscript text.

4) For more information: There is space at the end of each article to list relevant web links for further consultation by our readers. Could you identify some relevant ones and provide such information as well? Some examples are patient associations, relevant databases, OMIM/proteins/genes links, author's websites, etc...

5) As part of the EMBO Publications transparent editorial process initiative (see our Editorial at <http://embomolmed.embopress.org/content/2/9/329>), EMBO Molecular Medicine will publish online a Review Process File (RPF) to accompany accepted manuscripts. This file will be published in conjunction with your paper and will include the anonymous referee reports, your point-by-point response and all pertinent correspondence relating to the manuscript. Let us know whether you agree with the publication of the RPF and as here, if you want to remove or not any figures from it prior to publication. Please note that the Authors checklist will be published at the end of the RPF.

6) Please provide a point-by-point letter INCLUDING my comments as well as the reviewer's reports and your detailed responses (as Word file).

I look forward to reading a new revised version of your manuscript as soon as possible.

Yours sincerely,

Zeljko Durdevic

The authors performed the requested editorial changes.

We are pleased to inform you that your manuscript is accepted for publication and is now being sent to our publisher to be included in the next available issue of EMBO Molecular Medicine.

Corresponding Author Name: Francesca Buricchi

Manuscript Number: EMM-2021-14035